# Maturation and detoxification of synphilin-1 inclusion bodies regulated by sphingolipids

Xiuling Cao[1,2†], Xiang Wu[1†], Lei Zhao[2†], Ju Zheng[2,3], Xuejiao Jin[1], Xinxin Hao[2], Joris Winderickx[3], Shenkui Liu[1], Lihua Chen[2,4*], Beidong Liu[1,2*]

[1]State Key Laboratory of Subtropical Silviculture, Zhejiang A&F University, Hangzhou, China; [2]Department of Chemistry and Molecular Biology, University of Gothenburg, Gothenburg, Sweden; [3]Functional Biology, KU Leuven, Leuven, Belgium; [4]Guangzhou National Laboratory, Guangzhou, Guangdong, China

## eLife Assessment

By combining Synthetic Genetic Array (SGA) analysis with state-of-the-art imaging techniques, this study provides strong evidence that sphingolipid metabolism controls the maturation of Parkinson's disease-associated Synphilin-1 inclusion bodies (SY1 IBs) on the mitochondrial surface in a yeast model. The authors present **compelling** proof that perturbing the sphingolipid metabolic pathway leads to delayed SY1 IB maturation and enhanced SY1-triggered toxicity. Altogether, the authors show the **important** role of sphingolipid metabolism in the detoxification process of misfolded proteins by facilitating large IB formation on the mitochondrial outer membrane.

**Abstract** Due to proteostasis stress induced by aging or disease, misfolded proteins can form toxic intermediate species of aggregates and eventually mature into less toxic inclusion bodies (IBs). Here, using a yeast imaging-based screen, we identified 84 potential synphilin-1 (SY1) IB regulators and isolated the conserved sphingolipid metabolic components in the most enriched groups. Furthermore, we show that, in both yeast cells and mammalian cells, SY1 IBs are associated with mitochondria. Pharmacological inhibition of the sphingolipid metabolism pathway or knockout of its key genes results in a delayed IB maturation and increased SY1 cytotoxicity. We postulate that SY1 IB matures by association with the mitochondrion membrane, and that sphingolipids stimulate the maturation via their membrane-modulating function and thereby protecting cells from SY1 cytotoxicity. Our findings identify a conserved cellular component essential for IB maturation and suggest a mechanism by which cells may detoxify the pathogenic protein aggregates through forming mitochondrion-associated IBs.

## Introduction

Proteins are vulnerable macromolecules and are often endangered by misfolding and aggregation during the life of the cell due to destabilizing mutations, stress conditions or unique metabolic challenges that occur during aging or disease (*Buchberger et al., 2010*; *Hartl and Hayer-Hartl, 2009*). To cope with protein damage and in order to survive, the eukaryotic cell has evolved complex protein quality control systems that monitor and maintain the integrity of its proteome, including refolding, degradation, and sequestration (*Buchberger et al., 2010*; *Frydman, 2001*; *Hartl and Hayer-Hartl, 2009*; *Lim and Yue, 2015*). The refolding of stress-denatured proteins and the degradation of misfolded proteins are assisted by molecular chaperones (*Chen et al., 2011*; *Hartl and Hayer-Hartl,*

*For correspondence:
chen_lihua@gzlab.ac.cn (LC);
beidong.liu@cmb.gu.se (BL)

†These authors contributed equally to this work

Competing interest: The authors declare that no competing interests exist.

*2009*). In addition to refolding and degradation, cells have another cellular strategy to cope with an overload of aberrant proteins or toxic intermediate species of aggregates, namely sequestration into specialized quality control IBs (*Bagola and Sommer, 2008*; *Kaganovich et al., 2008*; *Kopito, 2000*).

It has become increasingly clear that the formation of IBs is like a 'triage' center, thus being protective machinery against proteostasis stress during aging or disease. Therefore, the inability to form IBs would poison the intracellular environment and cause cell dysfunction or even death (*Lim and Yue, 2015*). The toxicity of protein aggregates is more pronounced in neurons partly due to their unique cellular architecture. Indeed, mounting evidence has linked a number of neurodegenerative diseases including Alzheimer's disease (AD), Parkinson's disease (PD), Huntington's disease (HD) and amyotrophic lateral sclerosis (ALS), to the formation of misfolded proteins aggregates containing IBs (*Ross and Poirier, 2004*). Although IBs have been extensively studied, including their unique characteristics (*Kaganovich et al., 2008*; *Ogrodnik et al., 2014*; *Weisberg et al., 2012*) and the IB clearance machinery (*Moldavski et al., 2015*), the genes and cellular components that regulate the IB maturation remain poorly understood. In addition, we have little insight into the effects of IBs on cellular membrane-bound structures as IBs have been found to interact with membrane domains (*Büttner et al., 2008*).

SY1 contributes to inclusion formation and has been implicated in the pathogenesis of PD, an incurable, progressive neurodegenerative disorder. The biological functions of SY1 are not fully understood, and whether SY1 is cytotoxic or protective remains controversial. In aged yeast cells, SY1 expression reduced survival and triggered apoptosis and necrotic cell death in a Sir2 dependent manner (*Büttner et al., 2008*). However, SY1 exerted neuroprotective effects by inhibiting reactive oxygen species (ROS) production and apoptosis in an in vitro PD model (*Shishido et al., 2019*), and SY1 attenuated mutant LRRK2-induced PD-like phenotypes and played a neuronal protective role in cultured cells and *Drosophila* (*Liu et al., 2016*). Transgenic mice expressing SY1 caused pathological inclusions and degeneration of dopaminergic neurons in the substantia nigra (*Krenz et al., 2009*), and reduced motor skill learning and motor performance (*Jin et al., 2008*; *Nuber et al., 2010*). In contrast, SY1 overexpression in A53T human α-synuclein (α-Syn) transgenic mice reduced astrogliosis and neuronal degeneration, resulting in increased lifespan (*Smith et al., 2010*). In an aged A30P α-Syn transgenic mouse model, Casadei et al observed that overexpression of SY1 promoted clearance of soluble and misfolded α-Syn, but did not rescue the motor deficit phenotype (*Casadei et al., 2014*).

Given that a high-content imaging approach has been successfully conducted for SY1 (*Zhao et al., 2016*), it was possible to study on a genome-wide scale the cellular components that are involved in IB formation by using yeast as a model. Moreover, since SY1 binds phospholipids, membranes, lipid droplets and synaptic vesicles (*Ribeiro et al., 2002*; *Takahashi et al., 2006*) and given that the membrane-binding properties of α-Syn are thought to be relevant for its pathological activity, we speculated that also for SY1 the binding to different types of lipid structures might be related to its cytotoxicity. Here, we combined the SGA approach with a high-content imaging-based screen to survey the genetic mutants with altered SY1 aggregation types using a yeast genome-wide mutant library. We identified the conserved sphingolipid metabolic pathway as an essential component for the maturation of SY1 IBs and for the detoxification of SY1 aggregates. In the sphingolipid-rich endomembrane system, SY1 aggregates tightly interact with mitochondria. Disruption of sphingolipid synthesis prevented this association, resulting in small, dispersed aggregates and enhanced SY1-induced cytotoxicity, notably mitochondrial toxicity. Furthermore, pharmacological inhibition of the sphingolipid pathway or knockdown of SPTLC2 altered the morphology of SY1 IBs and increased their cytotoxicity in mammalian cells, and this could be reversed by increasing the level of dihydrosphingosine (DHS), a sphingolipid synthesis intermediate.

## Results
### SY1 IB management in yeast cells

The formation of SY1 IBs in the yeast (*Saccharomyces cerevisiae*) has been studied by expressing the protein (DsRed-SY1) with a plasmid pYX212 under the control of a constitutive *TPI1* promoter (*Zhao et al., 2016*). Using a similar approach, we observed different phenotypes of SY1 aggregates in yeast cells. During the lag phase (4 hr), multiple small IBs presented in the cytoplasm but in the exponential growth phase (16 hr), these small aggregates merged into larger IBs, resulting in a decreased number

of smaller aggregates. Subsequently, when the cells entered the stationary phase (48 hr), most of the SY1-expressing cells retained only one to a few large IBs (*Figure 1A*). Thus, consistent with previously reported data (*Büttner et al., 2008*), this time-course study demonstrated a maturation process of SY1 IBs in yeast cells. To study the details of this process, we defined the IB phenotypes as Class 1 (one aggregate in a cell), Class 2 (two aggregates in a cell), and Class 3 (three or more aggregates in a cell), respectively (*Figure 1B*), as previously reported for mutant huntingtin inclusions (*Yang et al., 2016*). We then monitored the SY1 IBs every 4 hr from 4 to 48 hr. Most of the SY1 IB-forming cells exhibited a Class 3 phenotype (96%) at the 4 hr time-point. During growth, the proportion of Class 3 ratio dropped over time, while the proportion of Class 1 and Class 2 ratios increased inversely. After 24 hr, the proportions became stable and Class 1 IBs accounted for approximately 50% when cells reached stationary phase (*Figure 1C*). All these observations supported the notion that there is a compartmentalization process of SY1 IBs in the yeast cell. We therefore hypothesized that the formation and maturation of IBs is a cytoprotective process that helps sequester more toxic aggregates/oligomers into larger inert deposits (*Kitamura et al., 2006*).

Dysregulation of the formation of pathogenic protein inclusions is the major hallmark of age-related neurodegenerative diseases. To investigate whether IBs formed by SY1 expression in old cells have the same maturation process as that in young cells. We isolated old cells and examined the SY1 aggregates in old cells (≥5 generations) and young cells (≤1 generation) (*Figure 1D*). In young cells, about 27% cells had SY1 inclusions while in the old cells the ratio increased to 61% (*Figure 1E*). Furthermore, when counted the number of aggregates in individual cells, we found that there were much more aggregates per cell in the old cell population (*Figure 1F*, 4.91±3.32 of old cells vs 2.64±1.29 of young cells). These findings suggested that there was a dysregulation of SY1 IB maturation in old cells. However, the underlying mechanism remains unclear. This motivated us to further investigate the genes and cellular components that regulate SY1 IB maturation.

## Genome-wide screening for cellular components regulating the maturation of SY1 inclusions

To dissect the genes and cellular components that mediate the maturation of IBs, a genome-wide, high-content imaging-based screening approach was performed (*Zhao et al., 2016*; *Figure 2A*). Prior to performing the screen, we determined the appropriate time-point to assess the phenotypes of SY1 IBs. As shown in *Figure 1A–C*, at 16–20 hr of growth ($OD_{600}$ is 3.0~4.0), approximately 50% SY1-expressing cells had Class 3 aggregates, and the ratios were stable thereafter. Therefore, we chose the time point 20 hr for the following screening and other studies. We set the same criteria as in the previous study (*Zhao et al., 2016*) to include a particular mutant strain as a potential hit if: (1) the difference in the number of cells with Class 3 SY1 IBs between wild-type and mutant cells is statistically significant ($p \leq 0.05$, $t$-test) and (2) the absolute difference of the percentage of wild-type and mutant cells carrying SY1 inclusions is at least 20%. Potential hits acquired from the screening were then confirmed by manual examination of the Class 3 inclusion phenotype using a microscopic approach.

The screen was performed three times independently. Based on the above criteria and subsequent manual confirmation, a list of 84 mutants that showed an increase in cells carrying Class 3 SY1 IBs was obtained (*Supplementary file 1*). GO enrichment analysis indicated that these mutants were enriched in the following modules: serine C-palmitoyltransferase (SPT) complex, SPOTS complex, protein urmylation, TORC2 complex, palmitoyltransferase complex, and others (*Figure 2B and C*). The complete list of GO terms can be found in *Supplementary file 2*. Among them, SPT complex and SPOTS complex, containing LCB1, LCB2, and TSC3 had the highest enrichment (*Supplementary file 2*, *Figure 2B and C*), corresponding to the mutants including *lcb1-4*, *lcb2-1*, *lcb2-2*, and *tsc3-2* in our screen. We performed gene complementation experiments on three mutants *lcb1-4*, *lcb2-1*, and *tsc3-2*. We performed a time course monitoring of aggregates in these mutants and found that there was no significant decrease in Class 3 aggregates in the mutants as in the wild type (*Figure 2—figure supplement 1*). Restoring the corresponding gene almost completely rescued the SY1 IB phenotype with significantly reduced Class 3 IBs (*Figure 2D and E*). Thus, our above screen suggested a link between the highly conserved sphingolipid metabolic pathway and the maturation of SY1 IBs. A substantial body of evidence has highlighted an important role of sphingolipid metabolism in several neurodegenerative diseases, including Alzheimer's disease (*Mielke and Lyketsos, 2010*), amyotrophic lateral sclerosis (*Cutler et al., 2002*), and Parkinson's disease (*France-Lanord et al., 1997*).

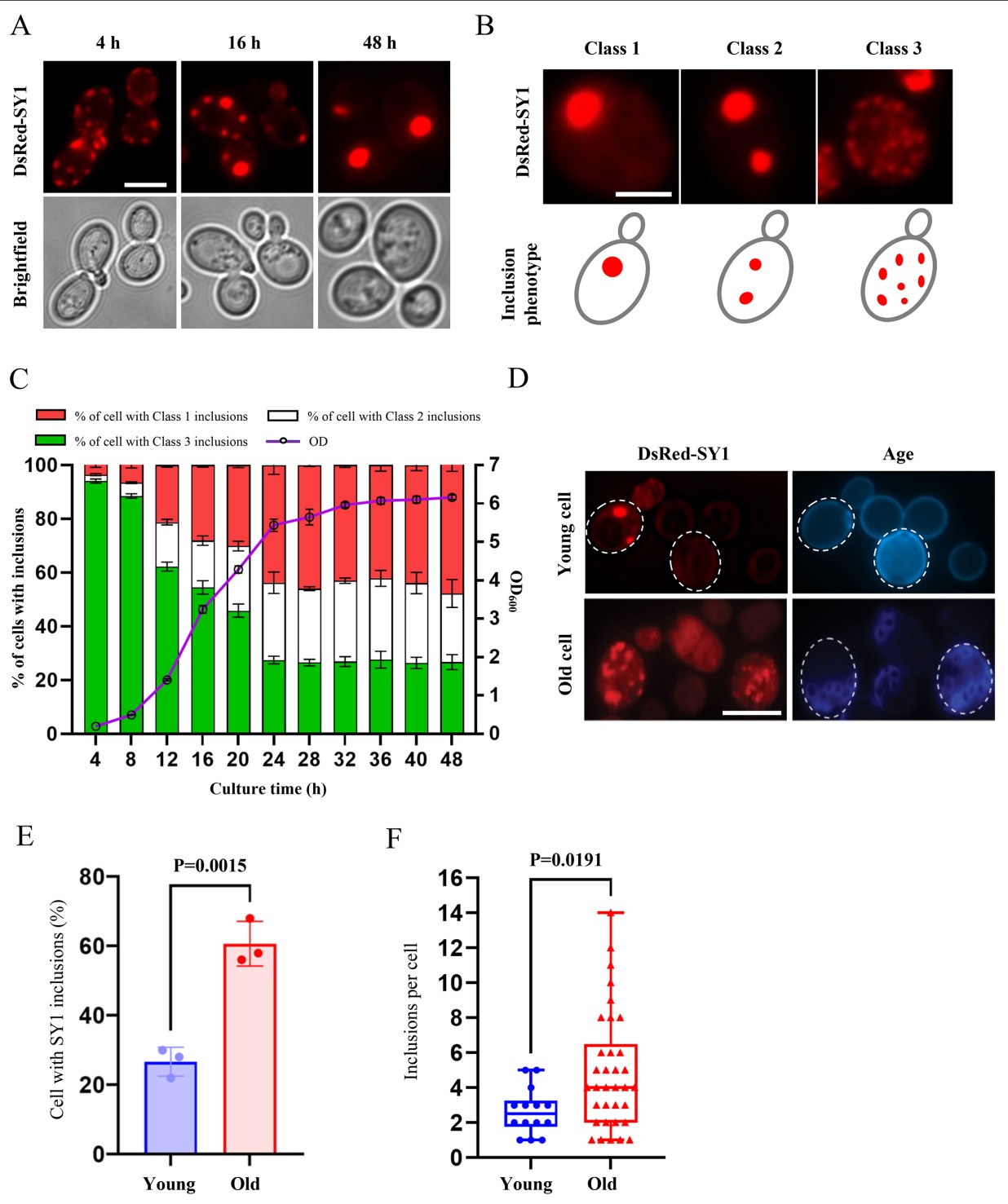

**Figure 1.** SY1 aggregates mature into large IBs over time in normal cells, but this process is defective in aged cells. (**A**) The maturation process of SY1 aggregates in yeast cells from scattered small aggregates to large single IBs over time. DsRed-SY1 was expressed in yeast with the constitutive promoter TPI1, the phenotype of SY1 aggregates was visualized at 4 hr, 16 hr and 48 hr. Scale bar = 2 μm. (**B**) Cells with different phenotypes of SY1 IBs are defined as three classes: Class 1 refers to only one IB in a cell, Class 2 refers to two IBs in a cell, and Class 3 indicates three or more IBs in a cell. Scale bar = 2 μm. (**C**) Time-course study showing that SY1 aggregates mature after 24 hr of cultivation, when the cells have traversed the diauxic shift to enter the respiratory and stationary growth phase. IBs were examined every 4 hr from 4 to 48 hr; green bars indicate percentage of Class 3; white bars indicate percentage of Class 2; red bars indicate percentage of Class 1; the purple curve indicates the optical density of the cell culture, indicated by the vertical axis on the right. Error bars indicate the SD, n=3 biological replicates. (**D**) Maturation of SY1 IBs was defective in old yeast cells. Scale bar =

*Figure 1 continued on next page*

*Figure 1 continued*

5 µm. (**E**) The proportion of cells with SY1 IBs was increased in old cells (61%) compared to young cells (27%). Data were analyzed using the Student's t-test. Error bars indicate the SD, n=3 biological replicates. (**F**) The number of SY1 IBs in a cell was increased in young cells (2.64±1.29) versus old cells (4.91±3.32). Data were analyzed using the Student's t-test. Error bars indicate the SD, Young n=14, Old n=34 biological replicates.

Therefore, we aimed to elucidate the functional role of sphingolipid metabolism in the regulation of SY1 IB maturation and cytotoxicity.

## Sphingolipids regulate SY1 IB maturation and cytotoxicity in yeast cells

The first step of intracellular sphingolipid de novo synthesis is identical in both yeast and humans, with palmitoyl-CoA and serine catalyzed by the SPT complex to form 3-Ketosphinganine, which could be blocked by myriocin (*Figure 3A*). 3-Ketosphinganine is subsequently catalyzed to dihydrosphingosine (DHS), and dihydroceramide, and further converted into complex sphingolipids. Since downregulation of SPT in yeast decreases sphingolipid levels (*Huang et al., 2012*), we first confirmed that exogenous addition of DHS in SPT mutants (*lcb1-4*, *lcb2-1*, and *tsc3-2*) could rescue the phenotype of increased Class 3 SY1 IBs (*Figure 3B and C*). In addition, blockade of intracellular sphingolipid metabolism pathways by myriocin strongly perturbed SY1 IB maturation process, resulting in elevated Class 3 IBs (*Figure 3D and E*). Again, addition of exogenous DHS could suppress the myriocin-induced Class 3 aggregates (*Figure 3D and E*). Next, we tested whether the SPT genes could regulate SY1 cytotoxicity in yeast cells. Expressing SY1 did not cause dramatic growth defect in wild-type cells, but led to severe growth defects in the SPT mutants (*Figure 3F*, left). Furthermore, this defect could also be rescued by the addition of DHS (*Figure 3F*, right). Taken together, these data suggest that sphingolipids may help cells reduce the cytotoxicity of protein aggregates by regulating the IB maturation process. These findings motivated us to further unravel the underlying mechanism by which sphingolipids regulate the maturation process and cytotoxicity of SY1 IBs.

## SY1 IBs associate with mitochondria in yeast cells

Sphingolipids are structural molecules of cell membranes that play an important role in maintaining barrier function and fluidity (*Bento-Oliveira et al., 2020*; *Hannun and Obeid, 2008*; *Schlarmann et al., 2021*). One of the primary modes of action of sphingolipids is to modulate membrane fluidity, thickness and curvature. In this context, we asked whether there is a direct association of SY1 IBs with certain membrane-bound organelles in the yeast cell. To address this question, we examined the location of SY1 IBs (20 hr) in cells and their possible association with membranous organelles visualized by corresponding marker proteins. Our results showed that SY1 IBs had no obvious association with the plasma membrane (Pma1-GFP), vacuole (Vph1-GFP), or ER (Sec63-GFP), but were strongly surrounded with mitochondria (Tom70-GFP) (*Figure 4A*, *Figure 4—figure supplement 1A*, about 80%). Since SY1 IBs have three different morphologies (Class 1–3) and the Class 3 IBs transition to Class 1 during maturation (*Figure 1A–C*), we wanted to confirm whether all these different IBs surrounded with mitochondria in a similar way. The results confirmed that all the three different IBs were equivalently associated with mitochondria (*Figure 4B*, *Figure 4—figure supplement 1B*). Furthermore, in myriocin-treated cells, most SY1 IBs were similarly surrounded by mitochondria (*Figure 4—figure supplement 1C*). Also, super-resolution microscopy and 3D-SIM techniques demonstrated that the SY1 IBs were surrounded by mitochondria (*Figure 4C*).

The next question is whether other pathogenic IBs are also associated with mitochondria. To address this question, we used the same expression system to overexpress TDP43, FUS1 or Htt103Qp (GFP-labeled) in yeast cells expressing Tom70-RFP and examined the possible association with mitochondria. Our results showed that the IBs of these proteins indeed interacted with mitochondria, similar to what we found for the IBs of SY1 (*Figure 4—figure supplement 2*). Hence, it appears that the association of IBs with mitochondria may be a common biological event among these disease proteins. To further study the consequences of this association and to decipher potential factors that regulate this association, we first investigated whether SY1 IBs physically associate with mitochondria. We performed four co-immunoprecipitation (Co-IP) experiments to determine whether SY1 could be detected in pull-downs for the inner mitochondrial membrane protein Tim44 or the outer mitochondrial membrane proteins Tom70, Tom37, and Sam35. The SY1 protein was indeed the case in extracts obtained from the Tom70-GFP, Tom37-GFP and Tim44-GFP strains (*Figure 4D*). Next, we performed

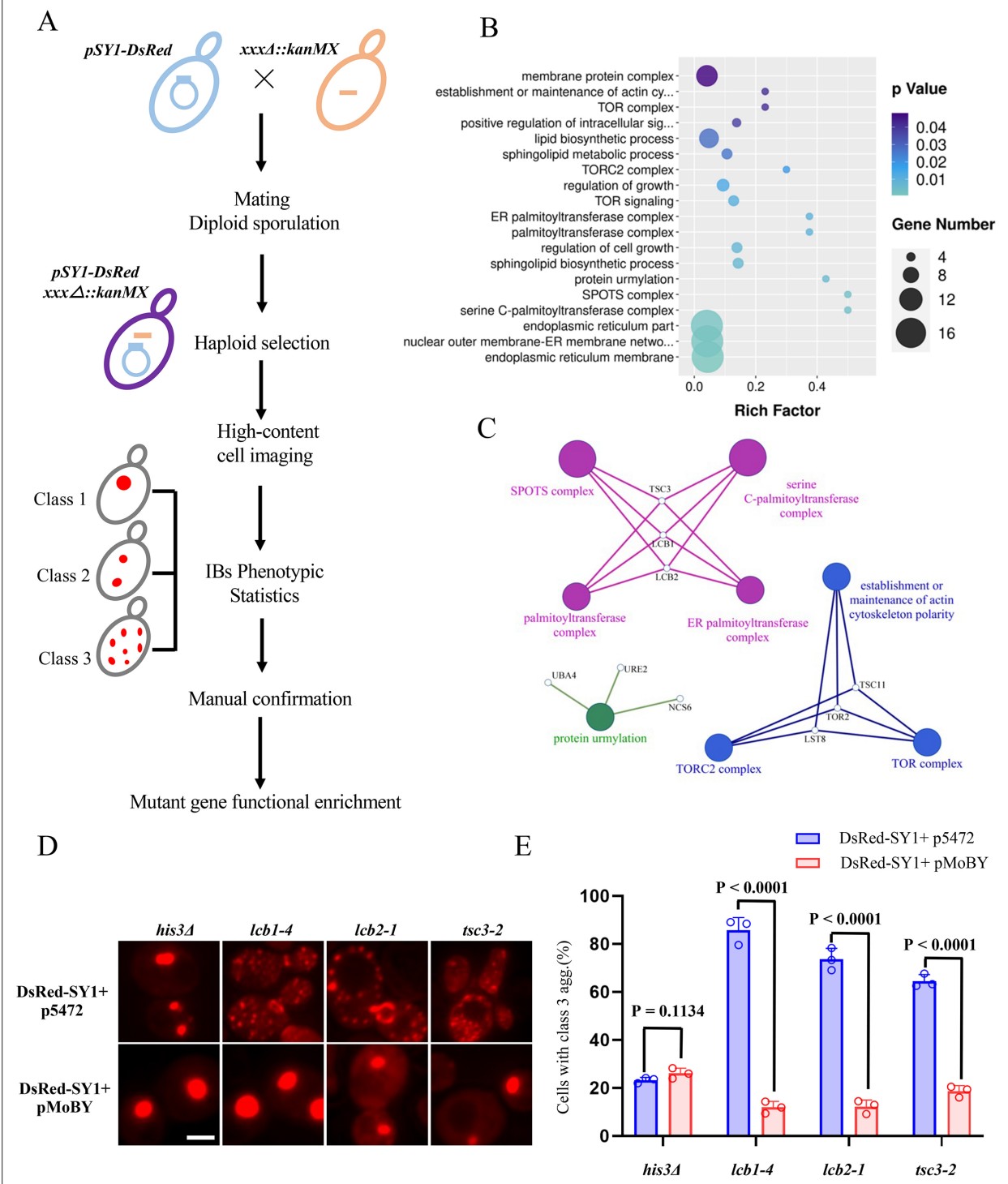

**Figure 2.** Genome-wide high-content imaging screen identifying the serine C-palmitoyltransferase complex (SPT) as a modulator of SY1 inclusion maturation. (**A**) Schematic workflow for identifying mutants that regulate SY1 inclusion maturation. (**B**) The GO enrichment analysis of genes whose mutants exhibited increased SY1 Class 3 aggregates (ClueGO, cut-off: p<0.05). (**C**) The network of the most enriched modules obtained above in 2B. (**D–E**) The increased SY1 Class 3 aggregates in the SPT mutant (*lcb1-4, lcb2-1, tsc3-2*) were rescued by backer-introducing the corresponding wild-type gene (a MoBY plasmid carries a WT gene, and p5472 is the empty vector control). Representative images from three independent experiments are shown in D, and quantifications were shown in E. Scale bar = 2 μm. Data were analyzed using the Student's t-test. Error bars indicate the SD, n=3 biological replicates.

The online version of this article includes the following figure supplement(s) for figure 2:

**Figure supplement 1.** The time course of SY1 IB maturation in the SPT mutants.

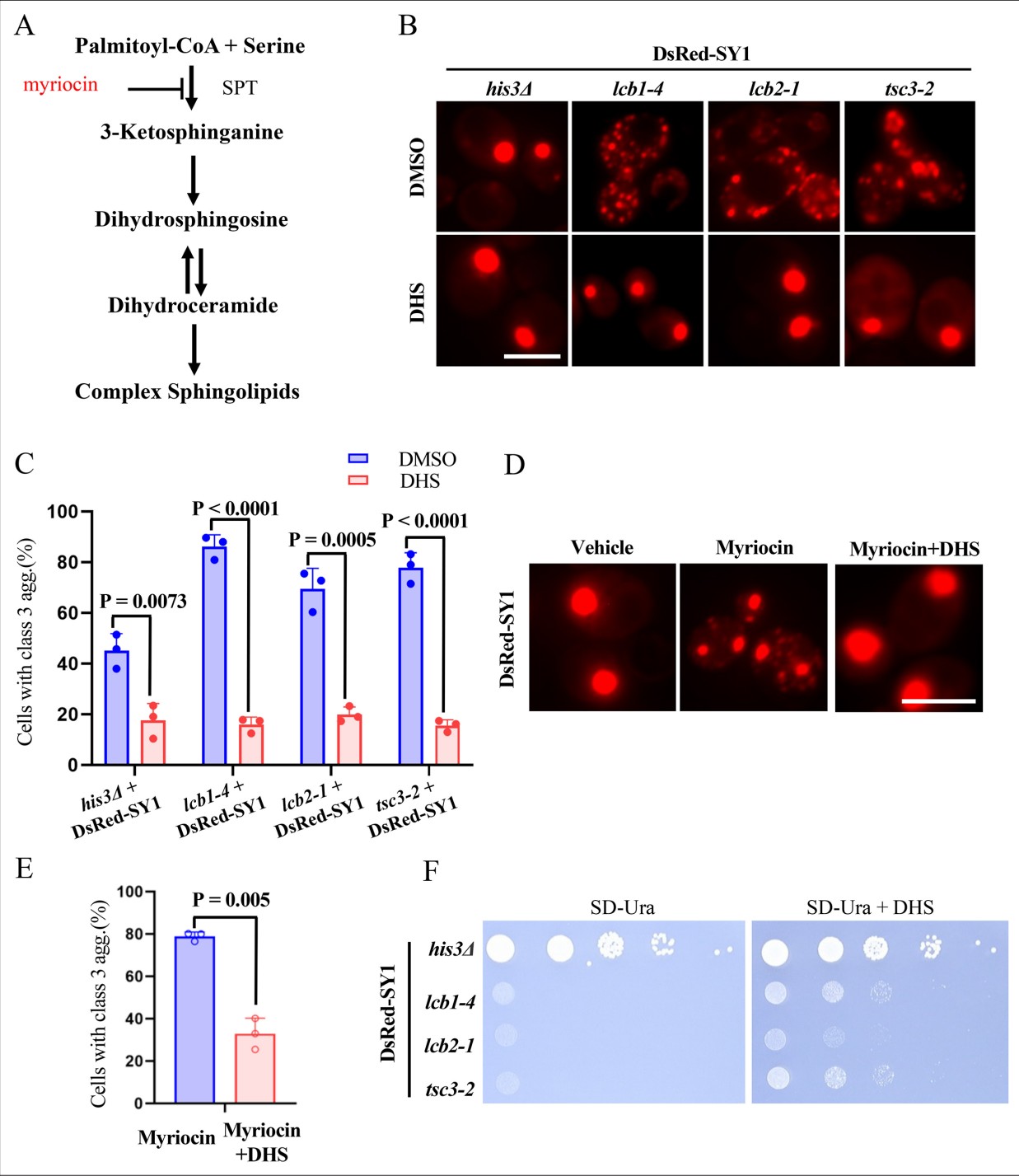

**Figure 3.** Sphingolipids regulate SY1 IB maturation and cytotoxicity in yeast cells. (**A**) A schematic diagram of the sphingolipid pathway in yeast cells. Myriocin blocks sphingolipid synthesis via inhibition of the serine C-palmitoyltransferase (SPT) complex. (**B–C**) Dihydrosphingosine (DHS) supplementation rescued the deficiency of SY1 IBs maturation in SPT mutants (*lcb1-4, lcb2-1, tsc3-2*). Representative images from three independent experiments are shown in B, and quantifications are shown in C. Scale bar = 2 μm. Data were analyzed using the Student's t-test. Error bars indicate the SD, n=3 biological replicates. (**D–E**) Blocking intracellular sphingolipid synthesis with myriocin resulted in defective maturation of SY1 IBs, and exogenous addition of DHS could reverse the effect of myriocin. Representative images from three independent experiments are shown in D, and quantifications are shown in E. Scale bar = 2 μm. Data were analyzed using the Student's t-test. Error bars indicate the SD, n=3 biological replicates. (**F**) DHS ameliorated the cytotoxicity of SY1 in SPT mutants (*lcb1-4, lcb2-1, tsc3-2*).

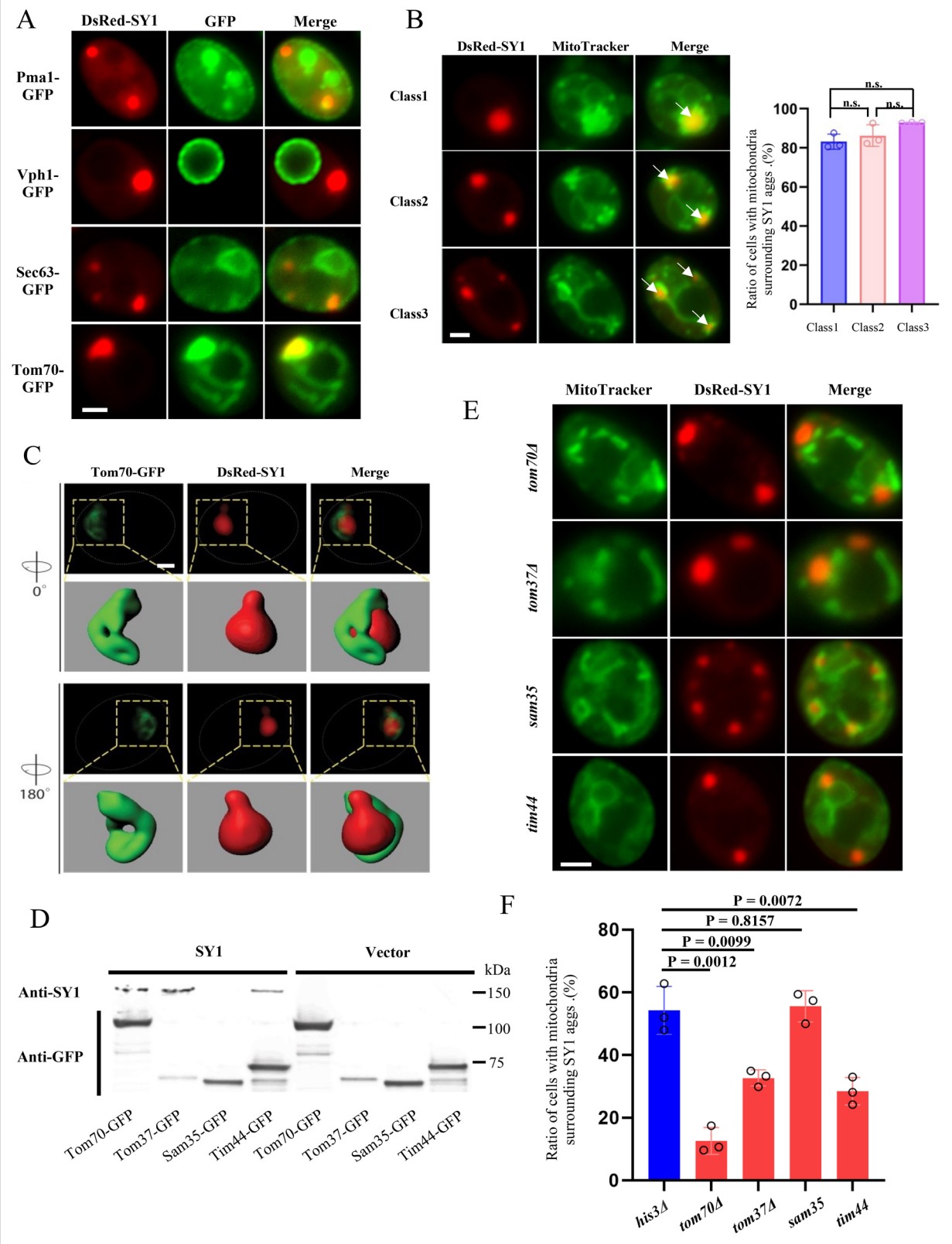

**Figure 4.** SY1 IBs associate with mitochondria in yeast cells. (**A**) SY1 IBs had no obvious association with the plasma membrane (Pma1-GFP), vacuoles (Vph1-GFP), or ER (Sec63-GFP), but were strongly associated with mitochondria (Tom70-GFP) (See also *Figure 4—figure supplement 1A*). Scale bar = 2 μm. (**B**) MitoTracker probe confirmed that all the three classes of IBs were closely associated with mitochondria (arrow heads indicate SY1 aggregates surrounded by mitochondria). Scale bar = 2 μm. n.s., not statistically significant. Data were analyzed using the Student's t-test. Error bars indicate the

*Figure 4 continued on next page*

*Figure 4 continued*

SD, n=3 biological replicates. (**C**) Super-resolution microscopy and 3D-SIM construction demonstrated that the SY1 IBs are packed and surrounded by mitochondria. Scale bar = 1 µm. (**D**) Co-immunoprecipitation assays showing that SY1 physically binds to several mitochondrial membrane proteins, including Tom70, Tom37, and Tim44. The yeast mitochondrial membrane proteins Tom70, Tom37, Sam35, and Tim44 were individually C-terminally tagged with GFP, and the strains were then transformed with a control plasmid (Vector) or a SY1-overproducing plasmid (SY1), respectively. Cells were grown for 24 hr and harvested for subsequent IP experiments. (**E–F**) The association of SY1 IBs with mitochondria decreases when the physical interaction between SY1 and mitochondrial membrane proteins is disrupted (*tom70Δ*, *tom37Δ*, *tim44*). Representative images from three independent experiments are shown in E and quantifications are shown in F. Scale bar = 2 µm. Data were analyzed using the Student's t-test. Error bars indicate the SD, n=3 biological replicates.

The online version of this article includes the following source data and figure supplement(s) for figure 4:

**Source data 1.** PDF file containing original western blot for *Figure 4D*, indicating the relevant bands.

**Source data 2.** Original file for western blot analysis displayed in *Figure 4D*.

**Figure supplement 1.** Statistics of SY1 IBs surrounded by cellular organelles.

**Figure supplement 2.** IBs of TDP43, FUS1, and Htt103Qp interact with mitochondria.

a microscopic analysis and found that the surrounding state of SY1 IBs with mitochondria was significantly hampered in *tom70Δ*, *tom37Δ*, and *tim44* mutants (*Figure 4E and F*), as well as an increased proportion of Class 3 aggregates in the mutants (*Figure 4—figure supplement 1D*). Furthermore, we expressed SY1 protein in BY4741 rho0 strain and found that the maturation and mitochondrial surrounding state of SY1 IB was not affected by mitochondrial activity (*Figure 4—figure supplement 1E and F*). Collectively, these results confirm that SY1 IBs are in physical contact with mitochondrial membrane components Tom70, Tom37, and Tim44, which is consistent with the fact that SY1 is a membrane lipid raft interacting protein (*Büttner et al., 2008*; *O'Farrell et al., 2001*; *Takahashi et al., 2006*).

## Sphingolipids regulate SY1-induced mitochondrial toxicity in yeast cells

Several lines of evidence have supported the notion that potentially toxic proteins such as α-Syn and TDP-43 can associate with mitochondria and damage the mitochondrial membrane, leading to cytotoxicity (*Bäuerlein et al., 2017*; *Busciglio et al., 2002*; *Hashimoto et al., 2003*; *Hsu et al., 2000*; *Park et al., 2020*; *Wang et al., 2019b*; *Wang et al., 2019a*). For example, pathogenic α-Syn aggregates preferentially bind to mitochondria, leading to decreased mitochondrial biogenesis (*Park et al., 2020*) and impaired cellular respiration (*Wang et al., 2019a*). To explore whether the interaction between SY1 IBs and mitochondrial membranes could impair mitochondrial function, we measured mitochondrial membrane potential and cell viability represented by ATP levels in SY1-expressing cells. The results showed that mitochondrial membrane potential and cell viability of SY1-expressing cells were significantly lower than that of control cells (*Figure 5A*). These changes indicate a decrease in mitochondrial function caused by SY1 IBs. Given that SY1 IBs interact with sphingolipid-enriched membrane raft domains and that sphingolipid metabolism regulates the maturation of SY1 IBs, we wondered whether sphingolipids play a role in the observed SY1-induced mitochondrial dysfunction. To address this question, we first blocked the sphingolipid synthesis by myriocin in the SY1-expressing yeast cells and measured mitochondrial membrane potential and cell viability. We found that myriocin treatment aggravated SY-1-induced cytotoxicity, as indicated by decreased mitochondrial membrane potential and cell viability (*Figure 5B*), without altering SY1 protein expression (*Figure 5—figure supplement 1A*). Second, we added exogenous DHS to SY1-expressing cells and observed ameliorated SY-1-induced cytotoxicity as indicated by increased mitochondrial membrane potential and cell viability (*Figure 5C*), without altering SY1 protein expression (*Figure 5—figure supplement 1B*). Notably, our assay showed that the knocking out of TOM70 or TOM37 reduced the mitochondrial toxicity caused by SY1 expression (*Figure 5—figure supplement 2*). Collectively, these results showed that sphingolipids regulated the mitochondrial toxicity induced by SY1 IBs, and suggested that sphingolipids might detoxify SY1 IBs through mitochondria.

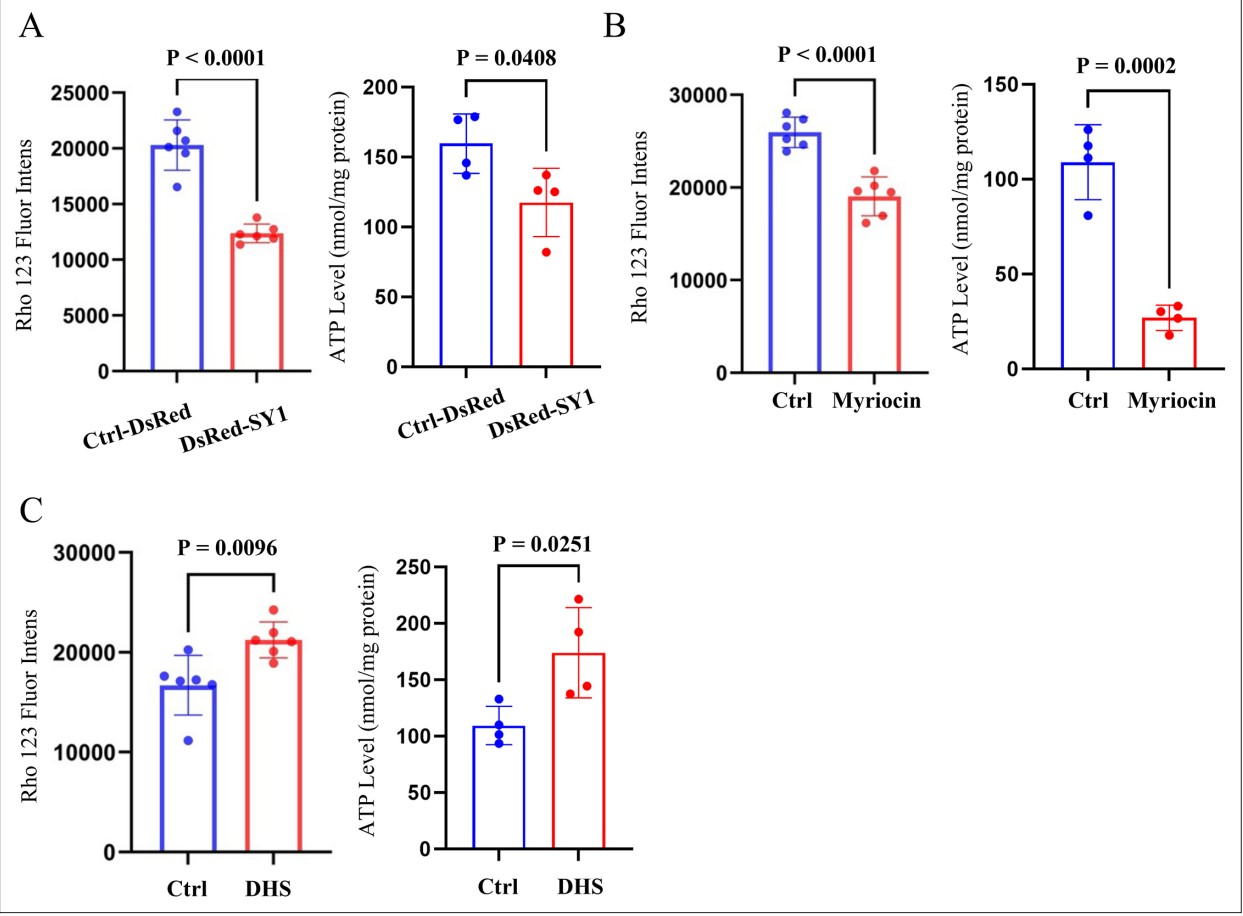

**Figure 5.** Sphingolipids regulate SY1-induced mitochondrial toxicity in yeast cells. (**A**) Overexpression of SY1 decreased mitochondrial membrane potential (indicated by Rho 123 fluor intensity) and reduced cell viability (indicated by ATP level) in yeast cells. Data were analyzed using the Student's t-test. Error bars indicate the SD, n=4–6 biological replicates. (**B**) Effects of myriocin treatment on SY1 cytotoxicity in yeast cells as judged by mitochondrial membrane potential and cell viability. Data were analyzed using the Student's t-test. Error bars indicate the SD, n=4–6 biological replicates. (**C**) Effects of exogenous addition of DHS on SY1 cytotoxicity in yeast cells based on mitochondrial membrane potential and cell viability. Data were analyzed using the Student's t-test. Error bars indicate the SD, n=4–6 biological replicates.

The online version of this article includes the following source data and figure supplement(s) for figure 5:

**Figure supplement 1.** The effect of myriocin (**A**) and DHS (**B**) treatment on SY1 protein expression in yeast cells.

**Figure supplement 1—source data 1.** PDF file containing original western blot for *Figure 5—figure supplement 1A and B*, indicating the relevant bands.

**Figure supplement 1—source data 2.** Original files for western blot analysis displayed in *Figure 5—figure supplement 1A and B*.

**Figure supplement 2.** SY1-induced mitochondrial toxicity decreased in *tom70Δ* and *tom37Δ* cells.

## SPTLC2 knockout alters the morphology of SY1 IBs and increases the cytotoxicity in mammalian cells

We then wondered whether the observed association with mitochondria and the cytotoxicity of SY1 IBs was conserved in mammalian cells. Similarly, SY1 forms IBs in mammalian cells, which are known as aggresomes (*Zaarur et al., 2008*). First, we expressed SY1 in HEK293t cells to study the association of SY1 IBs with mitochondria. A 3D structure between SY1 IB and mitochondria was constructed by fluorescence 3D reconstruction and it confirmed that in these mammalian cells the SY1 IBs are in close contact with mitochondria, although the morphology of the mitochondrion is different (*Figure 6A*). Furthermore, the mitochondrial membrane potential was significantly decreased in SY1-expressing cells, while the levels of mitochondrial superoxide were significantly increased (*Figure 6B*), indicating impaired mitochondrial function by SY1 IBs. As such, our data suggest that in yeast and mammalian

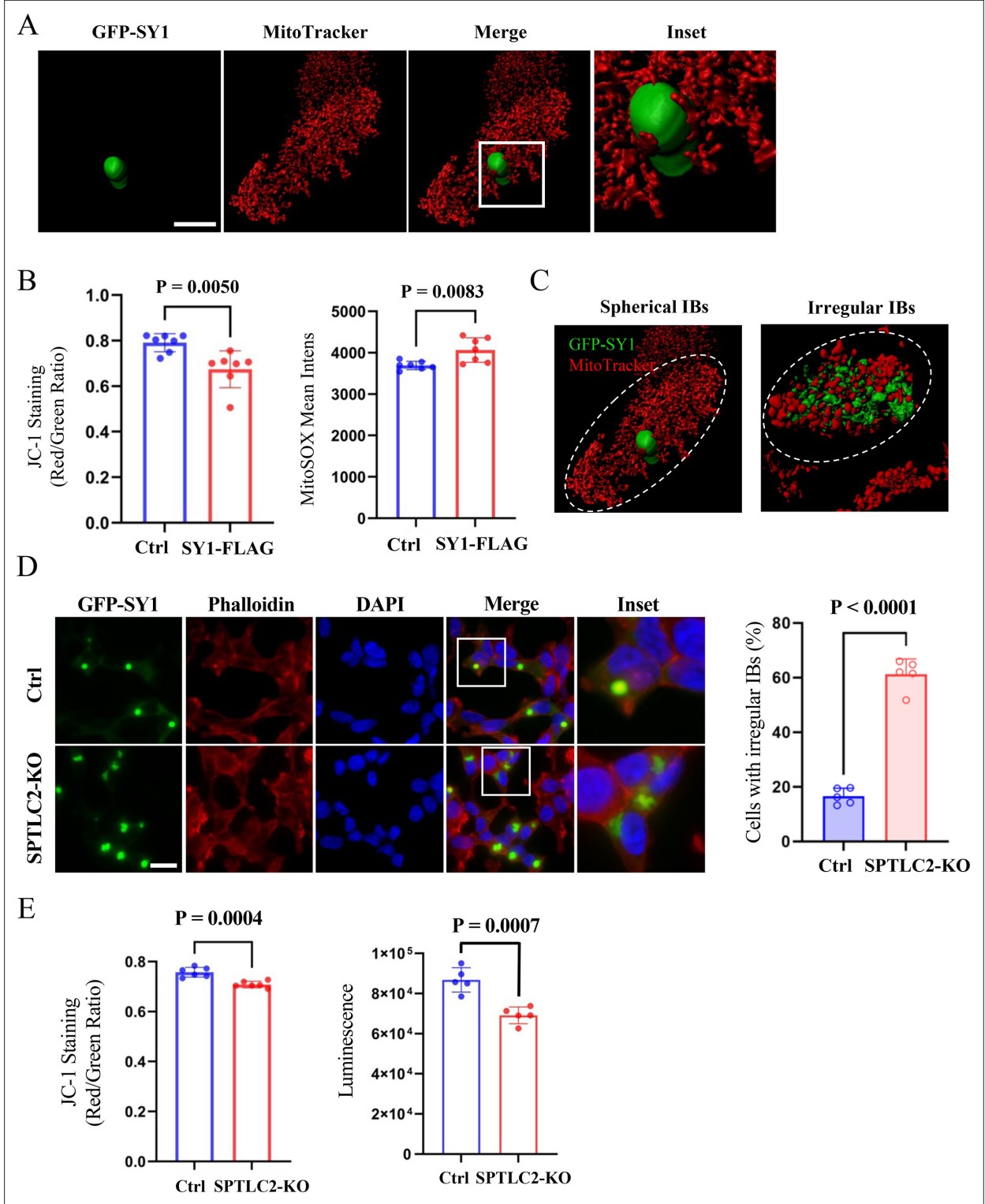

**Figure 6.** SPTLC2 knockout alters the morphology of SY1 IBs and increases the cytotoxicity in mammalian cells. (**A**) The 3D reconstruction showed that SY1 IBs were associated with mitochondria in mammalian HEK293t cells. GFP-SY1 was expressed in HEK293t cells and mitochondria were probed with MitoTracker. The 3D structure was constructed using Imaris software. Scale bar = 10 μm. (**B**) SY1 expression decreased mitochondrial function in HEK293t cells. Mitochondrial membrane potential was monitored by JC-1 staining, mitochondrial superoxide was analyzed by MitoSOX mean intensity values. Data were analyzed using the Student's t-test. Error bars indicate the SD, n=7 biological replicates. (**C**) The 3D structures of spherical and irregular SY1 IBs, as well as their interaction with mitochondria was visualized using Imaris software (see also *Figure 6—figure supplement 2*). (**D**) SPTLC2 knockout HEK293t cells had impaired management of SY1 IBs, with increased dispersed irregular IB structures. Scale bar = 20 μm. Data were

*Figure 6 continued on next page*

*Figure 6 continued*

analyzed using the Student's t-test. Error bars indicate the SD, n=5 biological replicates. (**E**) SPTLC2 knockout increased SY1 cytotoxicity. Mitochondrial membrane potential (indicated by JC-1 staining) assays / cell viability (indicated by luminescence) were analyzed. Data were analyzed using the Student's t-test. Error bars indicate the SD, n=5-6 biological replicates.

The online version of this article includes the following source data and figure supplement(s) for figure 6:

**Figure supplement 1.** SPTLC2-Knockout by CRISPR/Cas9 genome editing.

**Figure supplement 2.** Statistics of SY1 IB formation.

**Figure supplement 3.** The expression of SY1 protein.The expression of SY1 protein in HEK293t SPTLC2 knockout cells.

**Figure supplement 3—source data 1.** PDF file containing original western blot for *Figure 6—figure supplement 3*, indicating the relevant bands.

**Figure supplement 3—source data 2.** Original files for western blot analysis displayed in *Figure 6—figure supplement 3*.

cells the formation and processing of SY1 IBs occur in a similar manner, leading to mitochondrial dysfunction through influencing the mitochondrial membranes.

To further confirm this, we analyzed whether sphingolipids are indeed involved in the formation and cytotoxicity of SY1 IBs in mammalian cells as well. To this end, we used the CRISPR/Cas9 approach to knock out SPTLC2, the ortholog of yeast LCB2 and one of the subunits of the mammalian SPT complex (*Figure 6—figure supplement 1*). Notably, knockout of SPTLC2 significantly reduced the accumulation of intracellular sphingolipids, but had little effect on normal cell division and morphology (*Tafesse et al., 2015*). Microscopic analysis showed that the morphology of SY1 IBs formed in mammalian cells can be divided into two forms, one compact and spherical and the other more dispersed with an irregular structure as if many small IBs stuck together, without clear boundaries or uniform centers (*Figure 6C*). In wild-type cells, the SY1 IBs were predominantly spherical. However, in SPTLC2 knockout cells, the morphology of SY1 IBs was significantly altered, with an increased number of dispersed irregular IBs (*Figure 6D*) and an increased number of SY1 IBs per cell (*Figure 6—figure supplement 2*). This is consistent with the phenotypes observed in SPT mutant yeast cells (*Figure 3B*). We also performed SPTLC2 gene complementation experiments in knockout cells, which reduced the number of cells forming IBs and the percentage of dispersed irregular IBs (*Figure 6—figure supplement 2*). Furthermore, mitochondrial function, as judged from mitochondrial membrane potential and cell viability, was decreased in the SPTLC2-deficient cells (*Figure 6E*), while the SY1 protein expression was not altered (*Figure 6—figure supplement 3*). Taken together, these results further substantiate that SY1 IBs are processed and managed similarly in mammalian cells and in yeast cells and that the sphingolipid pathway plays a role in the cytotoxicity of SY1 IBs in mammalian cells as well.

## Sphingolipids regulate SY1-induced cytotoxicity and SY1 IB morphology in mammalian cells

Because the initial steps of sphingolipid biosynthesis are conserved from yeast to humans, myriocin also inhibits the de novo synthesis of intracellular sphingolipids in mammalian cells (*Ogretmen, 2018*). We added myriocin to SY1-expressing HEK293t cells, which significantly increased the number of SY1 IBs as the SPTLC2-deficient cells (*Figure 7—figure supplement 1A*), and then examined the corresponding changes in cytotoxicity. The results showed that inhibition of the intracellular sphingolipid synthesis pathway by myriocin enhanced SY1 cytotoxicity based on mitochondrial membrane potential and ATP content assays (*Figure 7A*). Furthermore, the addition of exogenous DHS rescued the cytotoxicity caused by SY1 expression (*Figure 7B*) without altering SY1 protein expression (*Figure 7—figure supplement 1B and C*).

We further examined the cytotoxicity of SY1 in SPTLC2 knockout cells with or without DHS. The results showed that DHS restored the morphology of IBs (*Figure 7C*), as well as the number of SY1 IBs formed in SPTLC2 knockout cells (*Figure 7—figure supplement 2*). Accordingly, DHS also ameliorated the SY1-instigated cytotoxicity by normalizing mitochondrial membrane potential and cell viability (*Figure 7D*), without affecting SY1 expression (*Figure 7—figure supplement 1D*). These results reinforce the notion that sphingolipid metabolism plays an important role in the formation of SY1 IBs and SY1-induced cytotoxicity.

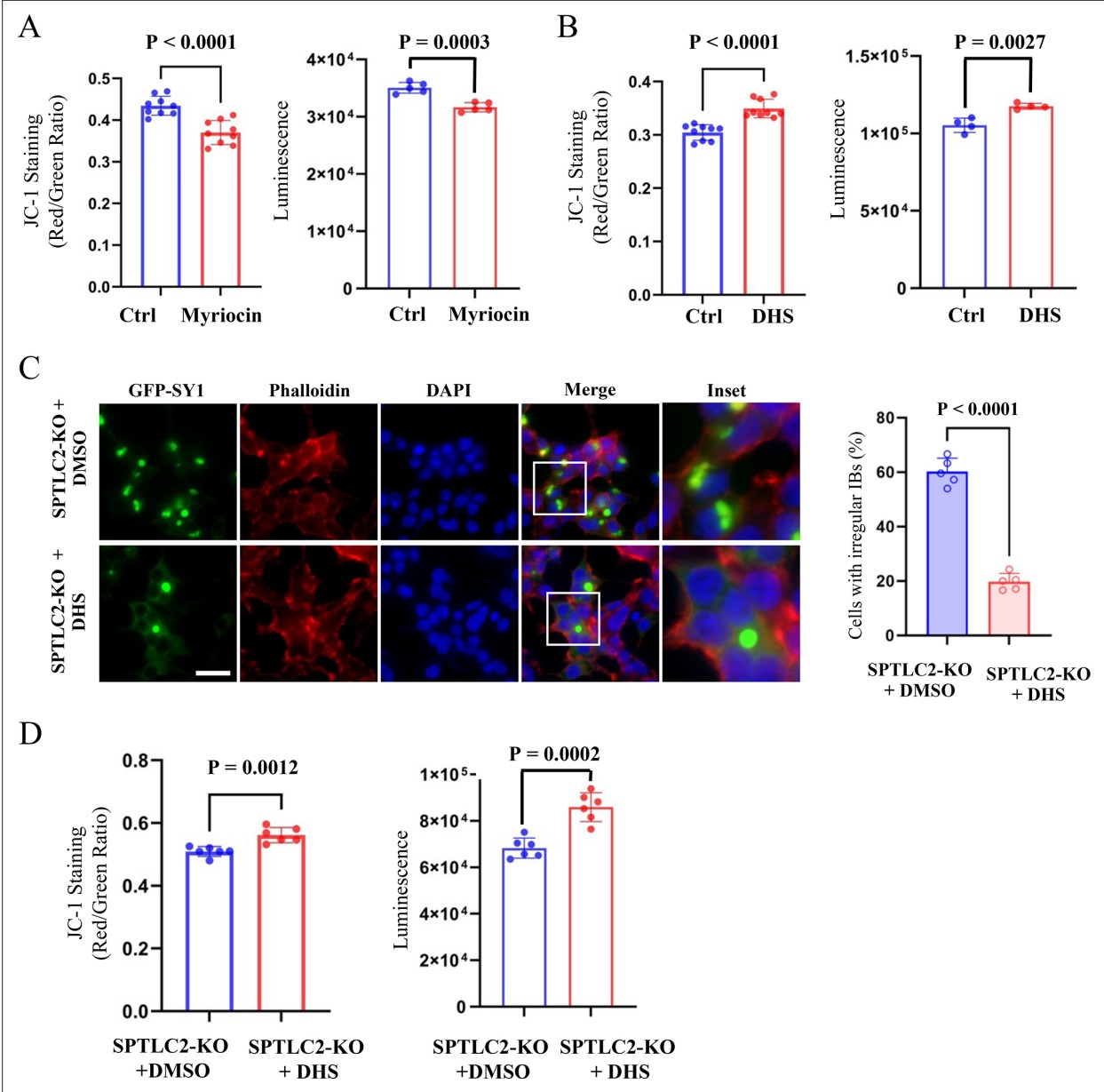

**Figure 7.** Sphingolipids regulate SY1-induced cytotoxicity and SY1 IB morphology in mammalian cells. (**A**) Inhibition of the intracellular sphingolipid synthesis pathway by myriocin enhanced SY1 cytotoxicity based on mitochondrial membrane potential and ATP content assays. Data were analyzed using the Student's t-test. Error bars indicate the SD, n=5-9 biological replicates. (**B**) Addition of exogenous DHS rescued the cytotoxicity caused by SY1 expression. Data were analyzed using the Student's t-test. Error bars indicate the SD, n=4-9 biological replicates. (**C**) Exogenous addition of DHS to SPTLC2 knockout cells restored the typical spherical SY1 IBs in HEK293t. Statistics of the ratio of cells with dispersed irregular IBs after exogenous addition of DMSO and DHS to HEK293t SPTLC2 knockout cells (See also *Figure 7—figure supplement 2*). Scale bar = 20 μm. Data were analyzed using the Student's t-test. Error bars indicate the SD, n=5 biological replicates. (**D**) DHS ameliorated SY1 cytotoxicity by normalizing mitochondrial membrane potential and cell viability. Data were analyzed using the Student's t-test. Error bars indicate the SD, n=6 biological replicates.

The online version of this article includes the following source data and figure supplement(s) for figure 7:

**Figure supplement 1.** The expression of SY1 protein in HEK293t and SPTLC2 knockout cells.

**Figure supplement 1—source data 1.** PDF file containing original western blot for *Figure 7—figure supplement 1B*, C and D, indicating the relevant bands.

**Figure supplement 1—source data 2.** Original files for western blot analysis displayed in *Figure 7—figure supplement 1B*, C and D.

**Figure supplement 2.** Statistics of SY1 IB formation.

## Discussion

In this study, we used a genome-wide high-content imaging-based screening approach to identify regulators of SY1 IBs formation. Among the 5500 mutants, 84 mutants exhibited increased Class 3 SY1 inclusions. GO enrichment analysis revealed the SPT complex as the most enriched functional group. Subsequent studies revealed that the sphingolipid pathway plays an important role in the processing of SY1 IBs and the detoxification of SY1 by mitochondria in yeast cells. Corresponding experiments in mammalian cells recapitulated the role of sphingolipids in the detoxification of SY1-induced mitochondrial toxicity. Taken together, our results suggest a role for sphingolipids in controlling the cytotoxicity of pathologically aggregated proteins in cells.

### IBs management and their cytotoxicity

The formation and accumulation of abnormal intracellular protein aggregates, including the pathological aggregates observed in neurodegenerative diseases, are generally considered to be a double-edged sword. On the one hand, their ability to co-aggregate and sequester basic cellular components, including molecular chaperones, causes cytotoxicity (*Gong et al., 2012*; *Olzscha et al., 2011*; *Park et al., 2013*). On the other hand, highly toxic, soluble oligomeric substances form protein aggregates can also protect cells (*Arrasate et al., 2004*; *Douglas et al., 2008*; *Tanaka et al., 2004*). Many studies have shown that converting small aggregates/oligomers into IBs can reduce the cytotoxicity of various aggregated proteins (*Miller et al., 2015*). In line with a previous report (*Büttner et al., 2008*), our current study demonstrated that the merging of smaller foci into IBs also occurred when SY1 was expressed in yeast and mammalian cells.

### Association of mitochondrial with protein aggregates

The formation of large intracellular IBs leads to severe impairment of membrane function of IB-bound organelles, thereby affecting organelle function. This has been confirmed in several experiments, such as the disruptive effect of Huntingtin polyQ IBs on ER function (*Bäuerlein et al., 2017*) and the effect of neuronal *C9orf72* poly-GA aggregates on the Proteasome (*Guo et al., 2018*). Several lines of evidence implicate mitochondrial dysfunction as a primary cause of PD. Environmental toxins such as mitochondrial complex inhibitors rotenone and 1-methyl-4-phenyl-1,2,3,6-tetrahydropyridine induce dopaminergic neuronal death (*Polito et al., 2016*). In addition, mitochondrial complex I deficiency and mitochondrial electron transport chain defects have been reported in the nigra of PD patients (*Haelterman et al., 2014*). However, it is unclear how these protein aggregates affect the mitochondria. As previously reported, small foci formed by SY1 started to merge into larger deposits when cells approached the diauxic shift to reprogram metabolism from fermentation to respiration (*Büttner et al., 2008*), which was confirmed by our present study (*Figure 1A*). In addition, we show that SY1 physically interacts with mitochondrial membrane proteins and that larger SY1 deposits are surrounded by mitochondrial structures in both yeast and mammalian cells (*Figures 4A–D , and 6A*). However, it is unknown why these IBs preferentially interact with mitochondria. This preference may be partly explained by a recent study in which the authors showed that the mitochondrial import receptor Tom70 could nucleate stress-induced aggregation of cytosolic proteins (*Liu et al., 2021*). In addition, impaired mitochondrial protein import could form cytosolic aggregates and thereby stimulate the aggregation of other proteins, including many pathogenic proteins such as α-Syn and amyloid β (*Nowicka et al., 2021*). In this study, however, our current study shows a fine management of aggregated proteins in eukaryotic cells—protein IBs are tightly associated with mitochondria once formed and continue to mature onto mitochondria.

In addition to SY1, there is a similar association of IBs and mitochondria with other aggregation-prone proteins, including TDP43, FUS1, and Htt103Qp (*Figure 4—figure supplement 2*), suggesting that mitochondrion-contact may be a common feature for the maturation of pathogenic protein IBs. Additional experiments are needed to confirm this hypothesis, although an interaction with mitochondria has already been described for the aforementioned and several other aggregation-prone proteins reviewed in *Abramov et al., 2017*. Furthermore, it would be of great value to illustrate how these IBs are sequestered to mitochondria and how the smaller IBs merge into large IBs with the help of mitochondria.

## The role of sphingolipids in SY1 IB maturation and mitochondrial function

Sphingolipids are structural molecules of cell membranes with important roles in maintaining barrier function and fluidity (*Hannun and Obeid, 2008*). In addition, bioactive sphingolipids have variety of functions that are involved in almost all major aspects of cell biology, and key sphingolipid metabolites are critical for normal brain development and function (*Hannun and Obeid, 2018*). Membranes and membrane dynamics are required for these processes and an imbalance of lipids, including ceramide and sphingolipid intermediates, may contribute to mitochondrial dysfunction that could lead to the demise of neurons (*Lin et al., 2019*). SY1 inclusions co-localize with lipid droplets and endomembranes, and SY1 interacts with cytoplasmic lipid rafts in mammalian cells and yeast (*Büttner et al., 2008*; *Takahashi et al., 2006*). The C-terminus of SY1 was found to bind selectively to acidic phospholipids, including phosphatidic acid, phosphatidylserine, and phosphatidylglycerol, but not to naturally charged phospholipids and the amino acid sequence 610–640 was found to represent the major determinant of phospholipid binding (*Takahashi et al., 2006*). The open question is how sphingolipids contribute to the SY1 IB maturation: do they influence this process directly, or indirectly by affecting other pathways? There was no significant enrichment for protein quality control or UPR-related pathways in our genome-wide screen, so it is unlikely that sphingolipids indirectly cause maturation of IBs by affecting these two pathways. This may require further investigation. Moreover, the relationship between sphingolipids and mitochondrial function during aging is not fully understood. It has long been appreciated that mitochondrial activity and quality decline with age reviewed in *Amorim et al., 2022*; *Lima et al., 2022*. On the other hand, sphingolipids play critical roles in the regulation of lifespan in yeast, worms, and flies, and in the regulation of senescence in mammalian cells (*Li and Kim, 2021*; *Singh and Li, 2018*; *Trayssac et al., 2018*). However, the changes in sphingolipid profiles during aging have not been fully characterized. In mammalian cells, there is an accumulation of ceramide and sphingosine during aging (*Sacket et al., 2009*), but the levels of other species including long-chain base sphingolipids are not reported. In this regard, additional studies will be needed to profile the sphingolipid species in the aged cells and to identify the major sphingolipids that regulate mitochondrial function.

## Materials and methods

### Key resources table

| Reagent type (species) or resource | Designation | Source or reference | Identifiers | Additional information |
|---|---|---|---|---|
| Antibody | Anti-Flag, rabbit monoclonal | Cell Signaling | D6W5B; RRID:AB_2572291 | 1:1000 |
| Antibody | Anti-GFP, rabbit polyclonal | Abcam | ab6556; RRID:AB_305564 | 1:1000 |
| Antibody | Anti-SY1, rabbit polyclonal | Affinity Biosciences | DF8619; RRID:AB_2841823 | 1:1000 |
| Antibody | Anti-PGK1, mouse monoclonal | Abcam | ab113687; RRID:AB_10861977 | 1:1000 |
| Antibody | Anti-GAPDH, rabbit polyclonal | Wanleibio | WL01114; RRID:AB_3665066 | 1:1000 |
| Antibody | Anti-rabbit IgG H&L (HRP), goat polyclonal | Abcam | ab6721; RRID:AB_955447 | 1:10000 |
| Antibody | Anti-mouse IgG H&L (HRP), goat polyclonal | Abcam | ab6789; RRID:AB_955439 | 1:10000 |
| Chemical compound, drug | myriocin | MCE | HY-N6798 | |
| Chemical compound, drug | dihydrosphingosine (DHS) | Sigma | D3314 | |
| Chemical compound, drug | sulfo-NHS-LC biotin | Thermo Fisher Scientific | 21335 | |
| Chemical compound, drug | streptavidin-conjugated paramagnetic beads | Thermo Fisher Scientific | 88817 | |
| Chemical compound, drug | MitoTracker Green | Invitrogen | M7514 | |

*Continued on next page*

*Continued*

| Reagent type (species) or resource | Designation | Source or reference | Identifiers | Additional information |
|---|---|---|---|---|
| Chemical compound, drug | Nonidet P-40 | Beyotime | ST2045 | |
| Chemical compound, drug | anti-GFP beads | chromoTek | GFP-Trap | |
| Chemical compound, drug | Laemmli buffer | Sigma | S3401 | |
| Chemical compound, drug | beta-mercaptoethanol | Thermo Fisher Scientific | 21985023 | |
| Chemical compound, drug | Rhodamine123 | Beyotime | C2007 | |
| Chemical compound, drug | MitoTracker Deep Red FM | Invitrogen | M22426 | |
| Chemical compound, drug | phalloidin-Atto 565 | Sigma | 94072 | |
| Commercial assay or kit | MitoSOX | Invitrogen | M36008 | |
| Commercial assay or kit | JC-1 | Beyotime | C2006 | |
| Commercial assay or kit | X-tremeGENE HP DNA Transfection Reagent | Roche | 6366236001 | |
| Commercial assay or kit | CellTiter-Lumi Luminescent Cell Viability Assay Kit | Beyotime | C0065S | |
| Commercial assay or kit | BCA Protein Quantification Kit | Beyotime | P0012S | |
| Cell line (*Homo-sapiens*) | HEK293t | NCACC | SCSP-502; RRID:CVCL_0063 | |
| Cell line (*Homo-sapiens*) | HEK293t SPTLC2 knockout cells | This paper | | |
| Strains, strain background (*S. cerevisiae*) | yeast single gene knockout collection (SGA-V2) | Charlie Boone Lab | | |
| Strains, strain background (*S. cerevisiae*) | essential gene temperature sensitive allele collection (ts-V5) | Charlie Boone Lab | | |
| Strain, strain background (*S. cerevisiae*) | BY4741 (MATa *his3Δ1 leu2Δ0 met15Δ0 ura3Δ0*) | Dharmacon Inc | YSC1048 | |
| Strain, strain background (*S. cerevisiae*) | Y7092 (MATα *can1Δ::STE2pr-his5 lyp1Δ ura3Δ0 leu2Δ0 his3Δ1 met15Δ0*) | **Tong and Boone, 2007** | | |
| Strains, strain background (*S. cerevisiae*) | Yeast GFP Collection | Invitrogen | 95702 | |
| Sequence-based reagent | Guide RNA for SPTLC2 knockout Guide RNA sequences #1: | This paper | Guide RNA | CGAATGGCTGCGTGGCGAAC |
| Sequence-based reagent | Guide RNA for SPTLC2 knockout Guide RNA sequences #2: | This paper | Guide RNA | AAGTACGGAACGGGTACGTG |
| Sequence-based reagent | Forward primer for SPTLC2 exon 1 sequence amplification: | This paper | PCR primers | GCCACCGCCTACAGAGCCTGC |
| Sequence-based reagent | Reverse primer for SPTLC2 exon 1 sequence amplification: | This paper | PCR primers | CCGGGAGTAAGACCTCCAGGCGC |
| Recombinant DNA reagent | pYX212-Dsred -SY1 | **Büttner et al., 2008** | | |
| Recombinant DNA reagent | pYX212-Dsred | **Büttner et al., 2008** | | |
| Recombinant DNA reagent | pYX212-Dsred -SY1-His | This paper | | Materials and methods section |
| Recombinant DNA reagent | pcDNA3.1-GFP-SY1 | This paper | | Materials and methods section |
| Recombinant DNA reagent | pcDNA3.1-SY1-Flag | This paper | | Materials and methods section |
| Recombinant DNA reagent | px459-SPTLC2-gRNA1 | This paper | | Materials and methods section |
| Recombinant DNA reagent | px459-SPTLC2-gRNA2 | This paper | | Materials and methods section |
| Software, algorithm | ImageJ | NIH | RRID:SCR_003070 | https://imagej.net/imagej-wiki-static/ |
| Software, algorithm | Imaris | Bitplane AG | RRID:SCR_007370 | https://imaris.oxinst.com |
| Software, algorithm | ImageXpress software | Molecular Devices | RRID:SCR_016654 | https://www.moleculardevices.com |
| Software, algorithm | ClueGO (V2.5.9) | **Bindea et al., 2009** | RRID:SCR_005748 | https://apps.cytoscape.org/apps/cluego |
| Software, algorithm | Cytoscape (V3.9.1) | Cytoscape | RRID:SCR_003032 | https://cytoscape.org/ |
| Software, algorithm | GraphPad Prism | GraphPad Software | RRID:SCR_002798 | https://www.graphpad.com/ |

## Plasmids

All plasmids used in this study are listed in *Supplementary file 3*. The plasmid pYX212-DsRed-SY1-His was constructed by replacing the URA3 cassette with the His3MX6 cassette by homologous recombination in yeast. The corresponding plasmids for complementation experiments were derived from the MoBY library (*Ho et al., 2009*). The plasmids in this collection were constructed based on the p5472 plasmid with its native promoter and terminator. To construct plasmid pcDNA3.1-GFP-SY1, the GFP fragment and the SY1 fragment were inserted into the Not I/Hind III digested pcDNA3.1 (-) vector using Exnase (Vazyme, C112). For the construction of plasmid pcDNA3.1-SY1-Flag, the PCR-expanded SY1-Flag fragment was inserted directly into the Not I/Hind III digested pcDNA3.1 (-) vector.

## Yeast strains and growth conditions

The yeast strains used in this study are listed in *Supplementary file 4*. Experiments were mainly performed in the BY4741 strain (MATa *his3Δ1 leu2Δ0 met15Δ0 ura3Δ0*) and its respective mutants. The *S. cerevisiae* MATα strain Y7092 (*Tong and Boone, 2007*) was used for query strain constructions. The yeast single gene knockout collection (SGA-V2) and the essential gene temperature sensitive allele collection (ts-V5) were kindly provided by Prof. Charlie Boone, University of Toronto, Canada. Strains expressing Pma1-GFP, Sec63-GFP, Vph1-GFP, Tom70-GFP, Tom37-GFP, Sam35-GFP, and Tim44-GFP were obtained from the yeast GFP collection (*Huh et al., 2003*). Yeast strains were grown in YPD (1% yeast extract, 2% peptone, and 2% glucose) or appropriate synthetic drop-out medium (0.17% yeast nitrogen base without amino acids and ammonium sulfate, 0.5% ammonium sulfate and the appropriate synthetic drop-out amino acid, and 2% glucose). Yeast cells were grown at 30 °C or at 22 °C in case of temperature sensitive (ts) alleles.

## SGA analysis

The experimental methods are based on a previous report (*Zhao et al., 2016*), and essentially consist of transforming the plasmids pYX212-Dsred-SY1 and pYX212-Dsred (control), respectively, into the yeast query strain Y7092. Subsequently, SGA mating was performed as previously described (*Tong et al., 2001*; *Tong et al., 2004*) to introduce the plasmids pYX212-Dsred-SY1 or pYX212-Dsred into the SGA-V2 and ts-V5 collections, respectively. All pinning steps for collection handling were performed using a SINGER ROTOR HDA robot (Singer Instrument, Watchet UK).

For the genome-wide high content imaging screening of SY1 IB morphology, the constructed collections expressing Dsred-SY1 were pre-cultured in 200 µL SD-Ura liquid medium in 96-well plates with antibiotics at 30 °C (22 °C for ts mutants) for 3 days without shaking. Then, each pre-culture sample was diluted to $OD_{600nm}$ 0.1 to a final volume of 200 µL of SD-Ura medium. After 20 hr (24 hr for ts mutants) of incubation with shaking at 30 °C (for both single and ts mutants), cells were fixed by adding formaldehyde to a final concentration of 3.7% for 30 min at room temperature and then washed twice with 1×PBS. For imaging, appropriate amounts of fixed cells were transferred to new 96-well glass bottom plates (Matri-plate) with 200 µL of 1×PBS in each well and imaged using ImageXpress MICRO (Molecular Devices Corporation (MDC), San Jose, USA), an automated cellular imaging system. To automatically quantify cells with different SY1 aggregation phenotypes, a customized subprogram of the MetaXpress software (MDC) was used (*Zhao et al., 2016*). Image preprocessing with shading correction and background subtraction was first performed; cells and IBs were separated from their background based on differences in signal intensity, respectively. Then, the number of IBs per cell was extracted by combining the two masks (cell and inclusion body). The percentages of the three types (class 1 to class 3) of cells were counted. All mutants that showed statistically significant differences as compared to the wild-type were re-streaked, individually tested, and further manually analyzed to confirm the phenotypic differences observed in the screen. At least 300 cells were counted for manual confirmation. For manual confirmation, cells were grown from a starting $OD_{600nm}$ of 0.1 in 15 ml Falcon tubes for 20 hr (24 hr for ts mutants) and fixed and washed as described above. All manual confirmations were verified using a Zeiss Axio Observer 7 microscope (Carl Zeiss, Jena, Germany).

## Spot assays

For spot assays, the mutants carrying DsRed-SY1 were cultured in 3 mL SD-Ura liquid medium at 22 °C for 24 hr. The strains were diluted to $OD_{600\ nm}$ = 0.5 and then serially diluted tenfold in sterile water in

96-well plates. The dilutions were spotted on plates with or without DHS. Images were captured after incubation at 30 °C for 2 days.

## Bioinformatic analysis

Candidate hits were analyzed for enrichment in the Gene Ontology (GO) using ClueGO (*Bindea et al., 2009*) in Cytoscape with a cut-off of p<0.05. Bubble and network plots of significantly altered GO terms and genes were generated using OmicStudio tools (https://www.omicstudio.cn/tool) and Cytoscape.

## Isolation of SY1-expressing old yeast cells

Old yeast cells overexpressing SY1 were obtained by magnetic sorting (*Smeal et al., 1996*). The wild-type BY4741 cells expressing Dsred-SY1 were cultured in 100 mL SD-Ura liquid medium, and then harvested after growth to 0.5 with an $OD_{600nm}$ of 0.1, washed three times in ice-cold PBS pH7.4, and incubated with 1 mg/mL sulfo-NHS-LC biotin labeling for 20 min at room temperature. Cells were further washed four times with cold PBS pH 8.0 and grown overnight in 100 mL new fresh medium. Cells were harvested the next day and washed three times with cold PBS pH 7.4, and incubated with 250 mL streptavidin-conjugated paramagnetic beads for 2 hr at 4 °C. Unlabeled cells were removed by magnetic sorting. The cells were then grown overnight in 100 mL fresh SD-Ura medium. The isolation process was repeated two more times. At the final isolation, cells were collected from the supernatant of the first wash and resuspended in PBS to serve as the young cells, while cells labeled with magnetic beads were centrifuged and resuspended in PBS to serve as old cells. The yeast bud scars were photographed through the DAPI channel using a Zeiss Axio Observer 7 fluorescence microscope to confirm and calculate the degree of yeast senescence. Subsequently, the SY1 aggregates were photographed through the Dsred channel to calculate the type of SY1 aggregates.

## Complementation of yeast mutants with MoBY plasmid

The mutants expressing pYX212-DsRed-SY1 were transformed with the corresponding MoBY plasmid or empty control vector (p5472). For the morphological observation experiment of SY1 IBs, the cells were cultured in SD-Ura, His liquid medium at 30 °C starting from $OD_{600nm}$ = 0.1 for 20 hr. The cells were then inspected by a fluorescence microscope and photographed.

## Myriocin treatment and DHS supplementation in yeast

Myriocin was dissolved in DMSO at a concentration of 10 mM. The stock solution was incubated in a 55 °C water bath for 15 min prior to use. DHS was prepared in distilled DMSO to 30 mM stock solution. For treatments, cells were grown overnight and diluted in medium to an initial $OD_{600nm}$ of 0.1 with myriocin or DHS at a final concentration of 25 or 20 μM, respectively.

## Microscopic analysis of SY1 IBs in yeast

For the microscopic analysis of SY1 IBs in yeast and the co-localization analyses of SY1 IBs with fluorescently labeled proteins Pma1-GFP, Sec63-GFP, Vph1-GFP, and Tom70-GFP, pYX212-DsRed-SY1 was transformed into corresponding cells and yeast strains were grown overnight and diluted in culture medium to a starting $OD_{600nm}$ of 0.1. After shaking at 30 °C for 20 hr, the cells were fixed by adding formaldehyde. Cell images were captured using Zeiss Axio Observer 7. For mitochondrial staining analysis, MitoTracker Green staining solution was prepared according to the manufacturer's protocol. After 20 hr of growth from an $OD_{600nm}$ of 0.1, $1\times10^7$ cells were fixed with formaldehyde for 30 min and then washed twice in 1×PBS (pH 7.4). The cells were resuspended in 1×PBS (pH 7.4) containing 125 nM MitoTracker probe and incubated for 30 min at room temperature. After one wash with PBS, the cells were imaged by microscopy.

## 3D-SIM microscopy

3D-SIM microscopy images were obtained as previously reported (*Song et al., 2014*). Briefly, yeast strain Tom70-GFP expressing pYX212-DsRed-SY1 was grown overnight and diluted in medium to a starting $OD_{600nm}$ of 0.1. After shaking at 30 °C for 20 hr, cells were fixed with 3.7% formaldehyde and washed twice with 1×PBS (pH 7.4). The ELYRA PS.1 LSM780 setup from Zeiss (Carl Zeiss, Jena, Germany) was used for 3D-SIM. Images were acquired with a 100×/1.46 Plan-Apochromat oil

immersion objective. Z-stacks with 100 nm spacing were used to scan the whole yeast in 3D-SIM. Zeiss Zen microscope software was used for acquisition, super-resolution processing and calculation, and 3D reconstruction. The ELYRA system was corrected for chromatic aberration in x-, y-, and z-directions using multi-color beads, and all obtained images were inspected and aligned accordingly.

## Co-IP for SY1 with mitochondrial membrane proteins in yeast

pYX212-DsRed-SY1 and pYX212-DsRed were transformed into GFP strains (Tom70-GFP, Tom37-GFP, Sam35-GFP, and Tim44-GFP) respectively. The strains were pre-cultured and diluted to 100 mL medium with $OD_{600nm}$=0.1. Cells were harvested when $OD_{600nm}$ reached 1.0 by centrifugation at 4000 rpm for 5 min at 4 °C and washed twice with 20 mL ice-cold $ddH_2O$ water. Cells were then washed with 1 mL ice-cold IP buffer (115 mM KCl, 5 mM NaCl, 2 mM $MgCl_2$, 1 mM EDTA, 20 mM HEPES/KOH, pH 7.5), and transferred to 1.5 mL Eppendorf tubes. Cell pellets were resuspended in 250 µL IP buffer containing protease inhibitors, DTT, and glass beads and disrupted using a FastPrep-24 device (MP Biomedicals). Cells were further lysed on ice for 20 min with additional IP buffer containing 0.5% Nonidet P-40. The lysates were centrifuged at 16,000 × $g$ for 20 min at 4 °C, and 700–800 µL of supernatant was transferred to a new tube. For the IP assay, 20 µL anti-GFP beads were resuspended in 100 µL IP buffer and mixed with the supernatant. The mixture was incubated for 1–1.5 h at 4 °C on an overhead rotator, centrifuged at 2000 rpm for 1 min, and washed four times with 1 mL IP buffer containing 0.5% NP-40. The final eluted proteins were boiled with 2 x Laemmli buffer / 5% beta-mercaptoethanol for 10 min at 95 °C and the supernatant was used for western blot analysis.

## Detection of mitochondrial membrane potential in yeast cells

To examine mitochondrial membrane potential in yeast cells, BY4741 yeast cells expressing pYX212-DsRed-SY1 or pYX212-DsRed were pre-cultured overnight and diluted to an $OD_{600nm}$ of 0.1. Cells were harvested at the endpoints, washed twice with 1×PBS (pH 7.4), and treated with Rhodamine123 at a final concentration of 10 µM. Cells were incubated for 20 min at 30 °C in the dark. Cells were then washed at least three times with 1×PBS (pH 7.4) before flow cytometric analysis using BD Accuri C6 (BD Biosciences, San Jose, USA) with FITC channel.

## Cell lines, culture condition, transfection, and drug treatment

The human embryonic kidney (HEK293t) cell line was obtained from the National Collection of Authenticated Cell Cultures (NCACC; Shanghai, China). The cell line was authenticated by STR profiling and tested negative for mycoplasma using a PCR-based assay. Cells were cultured in Dulbecco's modified Eagle's medium (DMEM; Gibco, c11995500bt) supplemented with 10% fetal bovine serum (FBS; Biological Industries, 04-001-1acs) and 1% penicillin-streptomycin (HycloneTM, SV30010) in a humidified incubator containing 5% $CO_2$ at 37 °C. Transfection was performed on six-well plates or confocal dishes for different purposes with plasmids using X-tremeGENE HP DNA Transfection Reagent according to the manufacturer's protocol. For drug treatment, cells were seeded at appropriate density in cell culture plates and cultured for 24 hr. Cells were treated with 25 µM myriocin for 3 days or 20 µM DHS for 2 days before subsequent analysis.

## Cell imaging in mammalian cells

To observe the association of SY1 IBs with mitochondria in mammalian cells, HEK293t cells were seeded into 35 mm confocal cell culture dishes (BioSharp), transfected with GFP-SY1, and incubated for 24 hr. The cells were washed with PBS and then stained with 10 µM MitoTracker Deep Red FM at 37 °C for 20 min. After staining, the cells were washed twice with PBS, and rinsed with PBS. Fluorescence images were captured using a Zeiss LSM 880 confocal microscope (Carl Zeiss, Jena, Germany), and Z-stack images for 3D rendering were taken at 0.4 µm interlayer distance. 3D rendering was performed using the Surface tool in the Imaris software.

For analysis of SY1 IBs formation under myriocin or DHS treatment, HEK293t cells were seeded in confocal culture dishes and transfected with pcDNA3.1-GFP-SY1 after treatment. Cells were fixed with 4% paraformaldehyde (Beyotime, P0099) for 20 min and permeabilized with 0.1% (v/v) Triton X-100 in PBS (Beyotime, P0096) for 15 min on ice. The cytoskeleton was then stained with phalloidin-Atto 565, followed by nuclei with DAPI (Southern Biotech, 0100–20). Images were captured using a Zeiss LSM 880 confocal microscope.

## Measurement of mitochondrial function in mammalian cells

HEK293t cells were detached and transferred to 75 mm flow tubes and exposed to 5 µM MitoSOX in serum-free medium for 30 min in a 37 °C cell incubator. Cells were then washed three times with HBSS buffer (Invitrogen,14025–092) and analyzed using a BD Accuri C6 flow cytometer.

Mitochondrial membrane potential (MMP) was measured with JC-1 according to the manufacturer's instructions. Briefly, treated cells were harvested, washed twice with PBS, and then stained in a mixture of 500 µL medium and 500 µL JC-1 at 37 °C for 20 min in the dark. Cells were washed three times with staining buffer and analyzed by flow cytometry. Changes in MMP are represented by the ratio of red to green fluorescence intensity.

## Measurement of cell viability and cytotoxicity

Cell viability was measured by monitoring the cellular ATP content using a CellTiter-Lumi Luminescent Cell Viability Assay Kit. Cells were treated and further incubated with 100 µL of detection reagent at room temperature for 2 min to promote cell lysis. Cells were further incubated at room temperature for 10 min to stabilize the luminescence signal, and ATP levels were then analyzed by using a Spark multimode microplate reader (Tecan Trading AG, Männedorf, Switzerland).

## Western blot analysis

Cells were harvested and lysed in RIPA buffer (Beyotime, P0013C) supplemented with a protease inhibitor of 1 mM PMSF (Beyotime, ST506). Total protein concentrations were then quantified using the BCA Protein Quantification Kit. A total of 20 µg of protein from each sample was separated by SDS-PAGE and transferred to nitrocellulose membranes (Cytiva, 10600001). The membranes were blocked with 1% BSA for 1 hr at 37 °C and incubated with primary antibodies: anti-Flag, anti-GFP, anti-SY1, anti-PGK1, anti-GAPDH, followed by HRP-conjugated secondary antibody incubation: goat anti-rabbit IgG. Protein bands were detected with ECL Prime Western Blotting Detection Reagent (GE, RPN2232) using an Amersham imager 600 (GE, Piscataway, USA).

## Generation of HEK293t SPTLC2 knockout cells by CRISPR/Cas9

For the generation of SPTLC2 knockout cell line, 2 guide RNA sequences for CRISPR were designed at CRISPR design web site (http://crispr.mit.edu/). Both guide RNAs target the exon 1 of the SPTLC2 gene (*Figure 6—figure supplement 1*). The DNA sequence of the guide RNA (gRNA) was synthesized and cloned into the pX459 CRISPR/Cas9-Puro vector to construct knockout plasmids px459-SPTLC2-gRNA1 and px459-SPTLC2-gRNA2.

Cells were seeded at appropriate densities in 35 mm dishes and grown for 24 hr before transfection with the plasmids. 48 hr after transfection, cells were treated with 2 µg/mL Puromycin (Sangon Biotech, A610593) for an additional 3 days. Single cells were then obtained by serial dilution and cultured for expansion. Genomic DNA was extracted from 18 single-cell expansion cultures and the exon 1 sequence of SPTLC2 was amplified by PCR. Two cell lines with deletion mutations were identified by sequencing (*Figure 6—figure supplement 1*).

## Statistical analysis

Statistical analyses were performed with GraphPad Prism 8.0 software. Data were analyzed using the Student's t-test and plotted as mean ± SD. Three or more biological replicates were analyzed unless explicitly stated. Image J was used to count cells and compare bands of immunoblot. Results were considered significant when $p < 0.05$. 'n.s.' stands for 'not statistically significant'.

## Acknowledgements

We thank Professor Charles Boone (University of Toronto) for providing SGA-V2 and ts-V5 strain collections, as well as the Moby plasmids. We also thank the Centre for Cellular Imaging at University of Gothenburg for microscopy support, and Center for Large-scale cell-based screening at University of Gothenburg for assistance with SGA analysis.

## Additional information

### Funding

| Funder | Grant reference number | Author |
|---|---|---|
| National Natural Science Foundation of China | 32000387 | Xiuling Cao |
| Natural Science Foundation of Zhejiang Province | LY23C060001 | Xuejiao Jin |
| Scientific Research Foundation of Zhejiang A and F University | 2021LFR053 | Xuejiao Jin |
| Fellowship from China Scholarship Council and KU Leuven | C14/17/063 | Ju Zheng |
| Cancerfonden | 19 0069 and 23 2769 Pj | Beidong Liu |
| Vetenskapsrådet | VR 2019-03604 and 2023-03560 | Beidong Liu |
| Basic and Applied Basic Research Foundation of Guangdong Province | 2023A1515010483 | Lihua Chen |
| FWO-Vlaanderen | G0C7222N | Joris Winderickx |
| KU Leuven | C14/21/095 | Joris Winderickx |

The funders had no role in study design, data collection and interpretation, or the decision to submit the work for publication.

### Author contributions

Xiuling Cao, Lihua Chen, Supervision, Funding acquisition, Investigation, Visualization, Writing – original draft, Writing – review and editing; Xiang Wu, Visualization, Methodology; Lei Zhao, Investigation, Visualization, Methodology, Writing – original draft; Ju Zheng, Funding acquisition, Methodology; Xuejiao Jin, Funding acquisition, Investigation; Xinxin Hao, Methodology; Joris Winderickx, Funding acquisition, Writing – review and editing; Shenkui Liu, Supervision; Beidong Liu, Conceptualization, Supervision, Funding acquisition, Investigation, Writing – review and editing

### Author ORCIDs

Xiuling Cao (iD) https://orcid.org/0000-0001-6940-4482
Joris Winderickx (iD) https://orcid.org/0000-0003-3133-7733
Lihua Chen (iD) https://orcid.org/0000-0003-3369-1875
Beidong Liu (iD) https://orcid.org/0000-0001-6052-8411

Reviewer #2 (Public review): https://doi.org/10.7554/eLife.92180.4.sa1
Author response https://doi.org/10.7554/eLife.92180.4.sa2

## Additional files

### Supplementary files

Supplementary file 1. Mutants with manually confirmed increased class 3 synphilin-1 aggregates.

Supplementary file 2. Distribution of GO terms among 84 candidate hits with significantly increased Class 3 aggregates via ClueGO analysis.

Supplementary file 3. Plasmids used in this study.

Supplementary file 4. Strains used in this study.

MDAR checklist

## Data availability

All data generated or analysed during this study are included in the manuscript and supporting files; source data files have been provided for *Figure 4*, *Figure 5—figure supplement 1*, *Figure 6—figure supplement 3*, and *Figure 7—figure supplement 1*.

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
