## [Editor Report · eLife Assessment]

By combining Synthetic Genetic Array (SGA) analysis with state-of-the-art imaging techniques, this study provides strong evidence that sphingolipid metabolism controls the maturation of Parkinson's disease-associated Synphilin-1 inclusion bodies (SY1 IBs) on the mitochondrial surface in a yeast model. The authors present **compelling** proof that perturbing the sphingolipid metabolic pathway leads to delayed SY1 IB maturation and enhanced SY1-triggered toxicity. Altogether, the authors show the **important** role of sphingolipid metabolism in the detoxification process of misfolded proteins by facilitating large IB formation on the mitochondrial outer membrane.

---

## [Referee Report · Reviewer #2 (Public review)]

Summary:

The authors used a yeast model for analyzing Parkinson's disease-associated synphilin-1 inclusion bodies (SY1 IBs). In this model system, large SY1 IBs are efficiently formed from smaller potentially more toxic SY1 aggregates. Using a genome-wide approach (synthetic genetic array, SGA, combined with a high content imaging approach), the authors identified the sphingolipid metabolic pathway as pivotal for SY1 IBs formation. Disturbances of this pathway increased SY1-triggered growth deficits, loss of mitochondrial membrane potential, increased production of reactive oxygen species (ROS), and decreased cellular ATP levels pointing to an increased energy crisis within affected cells. Notably, SY1 IBs were found to be surrounded by mitochondrial membranes using state-of-the-art super-resolution microscopy. Finally, the effects observed in the yeast for SY1 IBs turned out to be evolutionary conserved in mammalian cells. Thus, sphingolipid metabolism might play an important role in the detoxification of misfolded proteins by large IBs formation at the mitochondrial outer membrane.

Strengths:

• The SY1 IB yeast model is very suitable for the analysis of genes involved in IB formation.

• The genome-wide approach combining a synthetic genetic array (SGA) with a high content imaging approach is a compelling approach and enabled the reliable identification of novel genes. The authors tightly checked the output of the screen.

• The authors clearly showed, including a couple of control experiments, that the sphingolipid metabolic pathway is crucial for SY1 IB formation and cytotoxicity.

• The localization of SY1 IBs at mitochondrial membranes has been clearly demonstrated with state-of-the-art super-resolution microscopy and biochemical methods.

• Pharmacological manipulation of the sphingolipid pathway influenced mitochondrial function and cell survival.

• The authors have carefully redone critical experiments to avoid any misleading interpretation of data.

Weaknesses:

• It remains unclear how sphingolipids are involved in SY1 IB formation.

Comments on revisions: No further comments

---

## [Author Response]

The following is the authors’ response to the previous reviews.

**Reviewer #1:**
(1) I still think that the authors need to set the importance of the differences in aggregation in the context of toxicity arising from protein misfolding/aggregation. While the authors state the limitation in the response, and I agree that a single manuscript cannot complete a field of investigation I still think that this is an important point missing from this manuscript.

We thank the reviewer for the comments, we are working to address this issue and will elucidate in our future studies.

(2) I retain my reservations about the fluorescence intensity data shown for Rho123, DCF, Jc1, and MitoSox. The errors are much lower than what we typically achieve in biological experiments in our as well as our collaborator's lab. A glimpse at published literature would also support our statement. Specifically, RHO123 shows a large difference in errors between Figure 5 and Figure 5 Supplement 2. The point to note is that the absolute intensities do not vary between these figures, but the errors are the order of magnitude lower in the main figures. I, therefore, accept these figures in good faith without further interrogation.

We really value these comments from the reviewer and also do not want to cause any potential misleading interpretations of the data. We have therefore asked a more experienced author to redo all the experiments on the physiological indicators (Rho123, JC1 and MitoSox) that directly reflect mitochondrial function, and left out the DCF data. The new experimental data are in line with our previous results. We have clearly described these changes in the Results, Materials and Methods and Figure legends sections.

The new data from the redo experiments are: Rho123 fluorescence intensity data in Figure 5A, B and C; Figure 6B; JC1 staining in Figure 6E; JC1 staining in Figure 7A, B and D.